# Screening Unlearnable Examples via Iterative Self Regression

## Abstract

Deep neural networks are proven to be vulnerable to data poisoning attacks. Recently, a specific type of data poisoning attack known as availability attacks, has led to the failure of data utilization for model learning by adding imperceptible perturbations to images. Consequently, it is quite beneficial and challenging to detect poisoned samples, also known as Unlearnable Examples (UEs), from a mixed dataset. To tackle this problem, in this paper, we introduce a novel Iterative Self-Regression approach for identifying UEs within a mixed dataset. This method leverages the distinction between the inherent semantic mapping rules and shortcuts, without the need for any additional information. Our investigation reveals a critical observation: when training a classifier on a mixed dataset containing both UEs and clean data, the model tends to quickly adapt to the UEs compared to the clean data. Due to the accuracy gaps between training with clean/poisoned samples, we employ a model to misclassify clean samples while correctly identifying the poisoned ones for identifying tainted samples. Furthermore, we find that it is more effective to differentiate between clean and poisoned samples and build the Iterative Self Regression algorithm. With incorporated additional classes and iterative refinement, the model becomes more capable of differentiating between clean and poisoned samples. Extensive experiments demonstrate that our method outperforms state-of-the-art detection approaches across various types of attacks, datasets, and poisoning ratios, and it significantly reduces the Half Total Error Rate (HTER) in comparison to existing methods.

## 1 Introduction

The recent emphasis on data-centric AI (Zha et al., 2023) underscores the importance and effectiveness of improving the model's performance from the perspective of data pre-processing optimization instead of solely the model design. According to this viewpoint, it becomes critical to address the challenges tied to those data-centric issues, including data collection, data preprocessing, data quality maintenance, *etc.*, for learning-based methods. In fact, a considerable portion of the machine learning process has been dedicated to the data issue (Whang et al., 2023). These efforts have gradually guided researchers towards a consensus: high-quality data is critical for enabling more advanced machine-learning algorithms to reach their full potential, which poses the data on par with the approach itself. However, prevalent challenges persist, given that many real-world datasets are limited in size, dirty (Natarajan et al., 2013; Frénay & Verleysen, 2013), biased (Tommasi et al., 2017), and in some cases, even contaminated with malicious intents (Shafahi et al., 2018).

Data poisoning attacks (Huang et al., 2020; Chen et al., 2017; Geiping et al., 2021; Liu et al., 2020; Nguyen & Tran, 2021) further intensify these difficulties and reveal the urgent requirement for the attention of the data management community. The significance of this issue is rooted in the deliberate actions of attackers who maliciously manipulate data to undermine the accuracy of AI applications. Compared to the natural degradation of signals or shifts in features, these contaminations are more insidious and can result in greater damage, leading to a more obvious performance drop. In this work, we focus on a specific type of data poisoning attack termed availability attacks (Yu et al., 2022; Wu et al., 2023). These attacks add nearly invisible perturbations to the training data, which makes the trained models fail to obtain useful knowledge for reaching reasonable performance. This attack severely impacts the availability of data in the era of big data.

The widely employed dataset search engines (Brickley et al., 2019; Castelo et al., 2021) have heightened the risks of potential threats and misuse. Malicious dataset providers might release metadata to the public, which can be automatically discovered and propagated through search engines. Availability attacks, by introducing invisible perturbations within the $\ell_p$ norm to the original image, present a considerable challenge in distinguishing between clean and poisoned samples. The malicious dataset owner could even create mixed data by poisoning only portions of the data, which further makes the detection more challenging. Researches (Fowl et al., 2021; Huang et al., 2021) demonstrate that incorporating even a small proportion of clean samples into an unlearnable dataset leads to an increase in test accuracy. As a result, it is difficult for data users to determine whether the dataset is normal because the trained model appears to have reasonable performance. Training on these unlearnable examples may not fully leverage the model's potency, compromising testing performance and wasting computational resources and time.

In this paper, we introduce Iterative Self Regression (ISR), an effective detection technique tailored for detecting a wide range of visually imperceptible UEs. Our analysis reveals a critical insight: when a classifier is trained on a dataset that combines UEs with clean data, the model tends to adapt more rapidly to the UEs than to the clean data. This suggests that when the model is evaluated on previously unseen UEs, it often demonstrates superior accuracy compared to its performance on clean samples. Exploiting the accuracy disparities between training on clean and poisoned samples, we design a model to misclassify clean samples while accurately identifying the poisoned ones, facilitating the detection of tainted data. Furthermore, our findings lead us to the formulation of the Iterative Self Regression algorithm. With additional classes and iterative refinement, the proposed approach achieves improved performance in distinguishing these two types of samples.

Our contributions can be summarized as:

- In this paper, we address a critical issue in the era of deep learning: how to identify and filter harmful unlearnable data, screening samples that cannot be learned from. Accordingly, we introduce Iterative Self Regression (ISR), the first detection method that aims at identifying visually imperceptible unlearnable examples.

- ISR capitalizes on the observation that models trained on datasets blending UEs with clean data tend to adjust more swiftly to UEs, resulting in a discernible accuracy differential, to enable the identification of UEs. Specifically, ISR integrates additional classes and undergoes iterative refinement, enhancing its discrimination between clean and poisoned samples.

- Extensive experiments demonstrate the superior performance of our method over state-of-the-art detection approaches when confronted with various types of attacks, datasets, and poisoning ratios. When employing a detection-purification strategy, the results further emphasize the method's robustness in strengthening defenses against UEs.

## 2 RELATED WORK

### 2.1 DATA POISONING

Data poisoning (Biggio et al., 2012; Hong et al., 2020; Huang et al., 2020; Koh et al., 2022) is an increasingly recognized challenge in the modern machine learning ecosystem. Essentially, it involves the malicious modification of training data (often in a passive manner) to deliberately distort the behavior of a machine learning model. These data poisoning attacks manifest in multiple forms, ranging from targeted attacks aimed at particular categories to availability attacks seeking to undermine overall test accuracy. For example, Backdoor attacks (Chen et al., 2017; Doan et al., 2021; Gu et al., 2019; Nguyen & Tran, 2021) are characterized by the manipulation of training data instances. This allows attackers to control the target model's output utilizing a predetermined trigger. Label flipping attacks (Xiao et al., 2012) opt to switch training labels while leaving the data instances untouched.

While most of the literature focuses on the malicious use of poisoning attacks, availability attacks are employed as a protective measure against unauthorized model training. For example, Error-Minimizing (EM) (Huang et al., 2021) poisons introduce error minimization noise to prevent deep learning models from absorbing knowledge. Targeted Adversarial Poisoning (TAP) (Fowl et al.,

2021) uses targeted adversarial examples of pre-trained models for availability attacks. Robust Error-Minimizing (REM) (Fu et al., 2022) introduces adversarial training based on the EM method to generate robust unlearnable examples. Self-ensemble protection (SEP) (Chen et al., 2023) uses multiple model checkpoints' gradients to generate poison in a self-ensemble manner. Linear separable Synthetic Perturbation(LSP) (Yu et al., 2022) reveal that the perturbations of several existing availability attacks are (almost) linearly separable and propose to use synthetic shortcuts to perform availability attack. Recently, One Pixel Shortcut (OPS) (Wu et al., 2023) delves into the model's susceptibility to sparse poisons and augments its resistance to adversarial training.

## 2.2 Existing Defense against availability attacks

Defense against availability attacks can be mainly classified into two distinct categories: preprocessing and training-phase defenses. The method based on preprocessing eliminates the poison added by the attacker by preprocessing the data before training. Recently, Liu et al. (2023b) proposed to purify the poisoned data using grayscale transformation and JPEG compression (Marcellin et al., 2000). Dolatabadi et al. (2023) demonstrate the efficacy of diffusion models (Ho et al., 2020) in removing data protection perturbations. Training-phase defense methods are characterized by alterations in the training procedure to enable model robustness, even when exposed to poisoned data. Existing work tends to adopt adversarial training (Madry et al., 2018) as the countermeasure. However, this approach is not without its shortcomings – the substantial computational overhead and extended training durations often outweigh its benefits, and there exists a tangible risk of compromising the performance of models trained on clean datasets. Recently, adversarial augmentations (Qin et al., 2023) introduce an innovative technique of applying a spectrum of augmentations.

## 2.3 Detection of Backdoor attacks

A similar but different task compared to availability attack detection is backdoor attack detection. To date, some defense methods have been proposed to detect and mitigate backdoor attacks. Steinhardt et al. (2017) unveils a general defense against poisoning attacks, leveraging outlier or anomaly detection techniques. However, a significant limitation of their approach is the prerequisite of a clean, trusted dataset to effectively train the outlier detector. Addressing this constraint, Chen et al. (2018) introduces a pioneering methodology capable of detecting poisonous backdoors without the necessity of a verified and trusted dataset. This approach involves analyzing the neural network activations associated with the training data. It assesses whether the data is poisoned and identifies the specific data points that are affected if poisoning is detected. Tran et al. (2018) propose Spectral Signature defense that removes the data with the top $\epsilon$ eigenvalues. Wang et al. (2019) propose Neural Cleanse defense that first reverse-engineers a trigger by searching for patches that cause strong misclassification, then prunes neurons with large activations. Peri et al. (2020) capitalize on the observation that adversarial examples exhibit distinct feature distributions in higher layers of a neural network compared to their clean counterparts. They introduce a straightforward yet remarkably effective defense mechanism called Deep k-NN, which is designed for detecting and removing poisoned samples by leveraging these differences in feature distributions. Recently, Liu et al. (2023a) introduces TeCo, a technique anchored in sample corruption consistency for the precise detection of trigger samples during testing.

## 3 Methodology

### 3.1 Preliminary

**Unlearnable Examples.** In the era of big data, the internet and search engines bring about massive volumes of data that can be used for training deep models. However, there is a possibility of data contamination caused by availability attacks. That is, with small imperceptible perturbations, the data appears normal still in visual appearance but results in the failure of training deep networks with unsatisfactory testing accuracy. Here, these perturbed samples are termed 'unlearnable examples' (UEs). Following the existing unlearnable research (Huang et al., 2021), we focus on the image classification task in our paper. Given a clean dataset $\mathcal{D}_c = \{(x_i, y_i)\}_{i=1}^N$ comprising $N$ training samples, where $x \in \mathcal{X} \subset \mathbb{R}^d$ represents the image and $y \in \mathcal{Y} = \{0, \cdots, C-1\}$ represents its corresponding labels. We consider a scenario in which a classifier is trained on the contaminated

unlearnable data, represented as $f_\theta : \mathcal{X} \to \mathcal{Y}$. To corrupt the model training, the existing methods introduce perturbations to the clean images, resulting in an unlearnable dataset defined as:

$$\mathcal{D}_u = \{(x_i + \delta_i, y_i)\}_{i=1}^N, \tag{1}$$

where $\delta_i \in \Delta_\mathcal{D} \subset \mathbb{R}^d$, with $\Delta_\mathcal{D}$ representing the perturbation set for $\mathcal{D}_c$. The objective of unlearnability is to ensure that a classifier $f_\theta$ trained on $\mathcal{D}_u$ exhibits poor performance on testing datasets when using $\mathcal{D}_u$ in the inference stage.

**Defenses to Unlearnable Examples.** The training data used for model training might be gathered from online user channels with their regularly shared visual content like images and videos. Given that some images might be disturbed to be unlearnable examples, which are visually indistinguishable from normal ones, the dataset we collect might include a mix of clean data and unlearnable examples. It becomes critical to differentiate between these two types of data if we hope to train a highly effective model. That also implies the importance of identifying unlearnable examples from the training data. Once identified, the training data can be purified by removing those samples or restoring their corresponding clean versions by methods like 'DiffPure' (Dolatabadi et al., 2023) to improve the overall availability of the dataset.

**Unlearnable Examples Detection.** Given a mixed dataset $\mathcal{D} = \mathcal{D}_u \cup \mathcal{D}_c$, where $\mathcal{D}_u = \{(x_u^i, y_u^i)\}_{i=1}^{N_u}$ consists of $N_u$ unlearnable examples and $\mathcal{D}_c = \{(x_c^i, y_c^i)\}_{i=1}^{N_c}$ including $N_c$ clean samples, the detection of unlearnable examples within the mixed dataset $\mathcal{D}$ turns to learn a mapping $f$ appropriate for a binary classification problem, relative to the poison ratio $p = \frac{N_u}{N_u+N_c}$. Mathematically, the problem can be formulated as:

$$\min_{f(x)\in\{0,1\}} \sum_{i=1}^{N_u+N_c} |f(x) - I\{x \in \mathcal{D}_u\}|. \tag{2}$$

**UEs make faster learners.** Recent research indicates that UEs offer easily learnable features that are closely linked to labels, commonly recognized as shortcuts (Yu et al., 2022; Sandoval-Segura et al., 2022). A notable observation is that when training a classifier on a dataset merging UEs and clean data, the model adapts to the UEs more quickly than it does to the clean data (Huang et al., 2021; Fowl et al., 2021; Yu et al., 2022). Basically, when training a classifier $F(\cdot|\theta)$ on the mixed dataset $\mathcal{D} = \mathcal{D}_u \cup \mathcal{D}_c$, the optimization process (regarding the loss $\mathcal{L}$) is given by

$$\theta = \arg\min_\theta \sum_{i=1}^{N_u} \mathcal{L}(F(x_i + \delta_i|\theta), y_i) + \sum_{i=1}^{N_c} \mathcal{L}(F(x_i|\theta), y_i). \tag{3}$$

Since $\delta_i$ serves as a shortcut, the optimization process will prioritize minimizing the loss on UEs over clean data. This indicates that when evaluated on previously unseen UEs, which typically follow a similar distribution to the UEs in the training set (Liu et al., 2023b; Yu et al., 2022; Wu et al., 2023), the model often shows superior accuracy compared to its performance on clean data.

To validate this phenomenon, we consider a mixed dataset $\mathcal{D}$ that includes an equal number of UEs and clean data. Subsequently, we randomly sample 50% of the data from $\mathcal{D}$ for classifier training and evaluate on the remaining data. Figure 1 illustrates the testing accuracy for both the unseen UEs and the clean data throughout each epoch. Notably, for EM (Huang et al., 2021) and OPS (Wu et al., 2023), the accuracy for UEs increases near 100% within a handful of epochs, while accuracy improvement for the clean data is relatively slow.

This observation suggests that leveraging the distinction in learnability between UEs and clean data, especially through the introduction of an early stopping mechanism during training, can help to filter out potential inaccuracies labeled as clean data. In each iteration, we randomly select training data from the mixed dataset and evaluate on the remaining data. Through this iterative process, we can progressively distinguish the clean data from the rest. Nevertheless, when clean data is part of the optimization process as depicted in Eq. 4, the accuracy for such data might not necessarily remain at a low value. Consequently, if there is no significant gap in accuracy between UEs and clean data, the filtering-based detection approach may fail to obtain the desirable results.

## 3.2 KEY INTUITION

**It is simpler to distinguish between UEs and clean data.** To tackle the above-mentioned issues, we reveal a key insight: Distinguishing between clean and poisoned samples is demonstrated to be

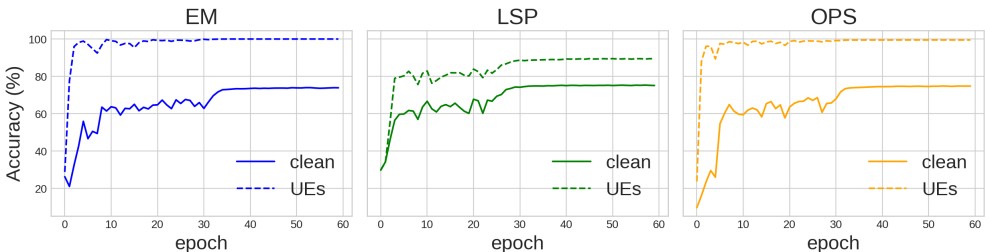

Figure 1: Test accuracy (%) on the unseen unlearnable data and clean data when training a classifier on a mixed dataset.

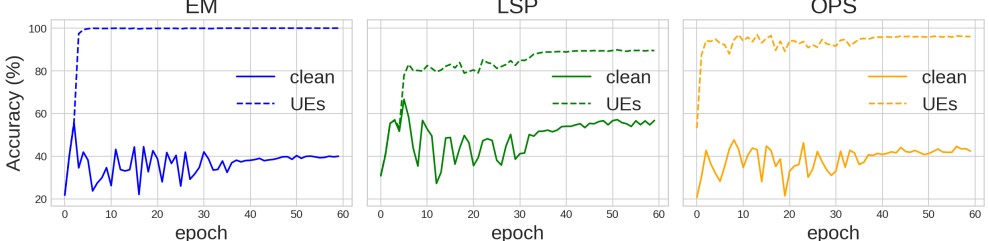

Figure 2: Test accuracy (%) on the unseen unlearnable data and clean data when training a classifier on a mixed dataset plus additional clean data with label set to $y + C$.

less challenging than guiding the model to misclassify clean samples while accurately classifying poisoned ones. In the filtering-based algorithm described above, the clean data, once filtered, is excluded from model training in subsequent iterations. Moreover, our studies provide evidence suggesting that these excluded clean data can considerably improve the effectiveness of detection.

Considering a mixed training dataset comprising 50% UEs and 50% clean data, each with their true labels $y$. Additionally, we have the excluded clean data, but in this case, we modify their labels to $y + C \in \{C, \cdots, 2C - 1\}$, where $C$ represents the number of classes in the dataset. These newly labeled data are then merged with the mixed dataset to form the dataset for the next iteration. The improved version of the optimization process is as follows:

$$\theta = \arg\min_{\theta} \sum_{i=1}^{N_u} \mathcal{L}(F(x_i + \delta_i|\theta), y_i) + \sum_{i=1}^{N_c} \mathcal{L}(F(x_i|\theta), y_i) + \sum_{i=1}^{N_c'} \mathcal{L}(F(x_i|\theta), y_i + C) \quad (4)$$

Subsequently, we randomly select 50% of the data for training, utilize the expanded label space $\{0, \cdots, 2C - 1\}$, and evaluate the remaining data with original labels $y \in \{0, \cdots, C - 1\}$. As illustrated in Figure 2, there is a noticeable decrease in the accuracy of clean data compared to previous experiments. This result is intuitive, as introducing clean data with additional class labels can lead to these clean data points being misclassified as $y + C$, thereby amplifying the difference between UEs and clean data. Moreover, as the number of iterations increases, the percentage of clean data with modified labels is expected to rise, likely resulting in more misclassifications as $y + C$, thereby further expanding the gap between UEs and clean data.

### 3.3 ITERATIVE SELF REGRESSION(ISR) FOR UNLEARNABLE EXAMPLES DETECTION

The Iterative Self Regression, as outlined in Algorithm 1, is designed to distinguish between unlearnable examples and clean samples, employing one subset for training and the other for testing and classification. Specifically, if a sample is incorrectly identified during testing, its class label is adjusted to $y + C$, expanding the label space for the entire dataset to $\{0, \cdots, 2C - 1\}$. Consequently, we introduce $C$ additional classes to the original classification model. In essence, our approach leverages the inherent responses of models when they encounter unlearnable examples. Through iterative training and testing, and by categorizing samples based on the model's prediction capabilities, we can effectively distinguish the unlearnable examples and clean samples within the dataset. During the iterative filtering process, poison samples tend to cluster within the initial $C$ categories, whereas clean samples are primarily categorized within the subsequent $C$ classes. However, in the early iterations, given a relatively low proportion of UEs in the training data, there exists a possibility that some UEs might be incorrectly identified as clean data during testing. As illustrated in Figure 3, as iterations increase, there is a corresponding rise in the FRR, indicating a growing

---

**Algorithm 1** Iterative Self Regression (ISR) for Unlearnable Examples Detection

---

**Input:** A mixed dataset $\mathcal{D}$ with $C$ classes, Classifier $F(\cdot|C_F)$ with $C_F$ classes, Percentage $p$, $N_{thre}$.
**Output:** Segregated sets of clean samples and unlearnable examples.

Initialize: $it = 0$, *retrieved* = False;

**while** $N_{update} > N_{thre}$ **do**
 # *filter clean data*
 Randomly divide $\mathcal{D}$ into two subsets: $\mathcal{D}_{train}^{(it)}$ and $\mathcal{D}_{val}^{(it)}$, where $\frac{|\mathcal{D}_{train}^{(it)}|}{|\mathcal{D}|} = p$;
 Randomly initialize and train a classifier $F(\cdot|2C)$ using $\mathcal{D}_{train}^{(it)}$ with early stopping;
 Evaluate $F(\cdot|2C)$ on $\mathcal{D}_{val}^{(it)}$, update $y_i \in \mathcal{D}_{val}^{(it)}$ by $y_i = y_i + C \cdot \boldsymbol{I}\{(\hat{y}_i = y_i) \,\&\, (y_i \in [0, C-1])\}$;
 Record the number of updated $y_i$ as $N_{update}$;
 # *retrieve unlearnable data*
 **if** $N_{update} > N_{thre}$ *& retrieved = False* **then**
  $\mathcal{D}_{val}^{(it)} = \{(x,y) \,|\, (x,y) \in \mathcal{D}, y \in [C, 2C-1]\}$;
  $\mathcal{D}_{train}^{(it)} = \mathcal{D} \backslash \mathcal{D}_{val}^{(it)}$;
  Randomly initialize and train a classifier $F(\cdot|C)$ using $\mathcal{D}_{train}^{(it)}$ with early stopping;
  Evaluate $F(\cdot|2C)$ on $\mathcal{D}_{val}^{(it)}$, update $y_i \in \mathcal{D}_{val}^{(it)}$ using $y_i = y_i - C \cdot \boldsymbol{I}\{\hat{y}_i = y_i - C\}$;
  *retrieved* = True;
  $N_{update} = 0$;
 **end**
 $it = it + 1$;
**end**
**Return** $\widehat{\mathcal{D}}_u = \{(x,y)|y \in [0, C-1]\}, \widehat{\mathcal{D}}_c = \{(x,y)|y \in [C, 2C-1]\}$

---

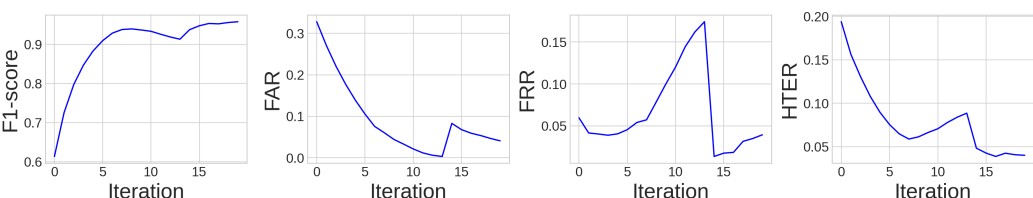

Figure 3: Performance Vs. Iterations on detecting EM-based UEs with 80% poison rate.

number of poisoned samples being misclassified with each subsequent iteration. To address this, we propose the idea of conducting retrieval once after several iterations. Furthermore, we establish a stopping criterion based on the number of correctly classified clean data during each iteration. Conventionally, we set the threshold $N_{thre}$ to be 2% of the total number of the entire dataset.

## 4 EXPERIMENTS

### 4.1 EXPERIMENTAL SETUP

**Datasets and models.** We use three image classification datasets, CIFAR-10 (Krizhevsky et al., 2009), CIFAR-100 (Krizhevsky et al., 2009), and 100-class subset of ImageNet (Deng et al., 2009) in our experiments. We implement ResNet-18 (He et al., 2016) as the image classification model.

**Methods for Generating UEs.** We explore a variety of representative methods to generate UE: EM (Huang et al., 2021), REM (Fu et al., 2022) with a perturbation bound of $\ell_\infty = 8$, AR (Sandoval-Segura et al., 2022), LSP (Yu et al., 2022) constrained by $\ell_2 = 1.0$, and OPS (Wu et al., 2023) with a perturbation bound of $\ell_0 = 1$.

**Competing Methods.** As our proposed ISR is the first UEs detection method without any baselines/competitors, we re-implement several state-of-the-art defensive techniques from backdoor at-

Table 1: Detection performance on CIFAR-10 dataset.

| Dataset | Ratio | Poisons | Deep k-NN (Peri et al., 2020) | | | | TeCo (Liu et al., 2023a) | | | | Our Method | | | |
|---|---|---|---|---|---|---|---|---|---|---|---|---|---|---|
| | | | F1-score↑ | FAR↓ | FRR↓ | HTER↓ | F1-score↑ | FAR↓ | FRR↓ | HTER↓ | F1-score↑ | FAR↓ | FRR↓ | HTER↓ |
| CIFAR-10 | 0.2 | EM | 47.68 | 3.82 | 73.39 | 38.60 | 49.55 | 19.53 | 79.55 | 49.54 | 83.29 | 5.65 | 17.13 | 11.39 |
| | | OPS | 46.32 | 2.26 | 73.43 | 37.84 | 51.50 | 19.90 | 79.90 | 49.90 | 91.06 | 4.66 | 9.52 | 7.09 |
| | | LSP | 37.95 | 16.52 | 78.29 | 47.41 | 49.51 | 19.63 | 79.65 | 49.64 | 79.27 | 60.67 | 19.32 | 39.99 |
| | | REM | 38.19 | 16.33 | 78.42 | 47.37 | 48.49 | 20.22 | 80.21 | 50.21 | 79.36 | 72.56 | 19.89 | 46.22 |
| | | AR | 38.57 | 19.31 | 79.61 | 49.46 | 51.43 | 19.66 | 79.64 | 49.65 | 78.66 | 78.00 | 19.95 | 48.97 |
| | 0.4 | EM | 69.85 | 3.26 | 42.89 | 23.08 | 49.51 | 40.25 | 60.24 | 50.25 | 88.88 | 0.10 | 15.60 | 7.85 |
| | | OPS | 61.01 | 21.46 | 49.40 | 35.43 | 48.50 | 40.54 | 60.44 | 50.49 | 92.96 | 0.55 | 10.02 | 5.41 |
| | | LSP | 64.21 | 12.12 | 46.72 | 29.92 | 49.69 | 39.53 | 59.60 | 49.57 | 80.23 | 2.97 | 24.37 | 13.67 |
| | | REM | 57.23 | 27.03 | 52.31 | 39.67 | 49.51 | 39.93 | 59.94 | 49.94 | 82.20 | 4.59 | 22.06 | 13.32 |
| | | AR | 48.10 | 36.80 | 58.3 | 47.55 | 48.79 | 40.14 | 60.11 | 50.13 | 60.03 | 42.96 | 39.95 | 41.46 |
| | 0.6 | EM | 74.44 | 29.52 | 23.34 | 26.44 | 51.10 | 59.64 | 39.69 | 49.67 | 95.98 | 4.06 | 3.93 | 4.00 |
| | | OPS | 70.79 | 32.44 | 6.22 | 19.33 | 50.26 | 60.01 | 40.01 | 50.01 | 95.25 | 2.37 | 8.03 | 5.20 |
| | | LSP | 60.00 | 50.45 | 30.57 | 40.51 | 49.70 | 61.37 | 41.13 | 51.25 | 93.96 | 4.29 | 8.54 | 6.42 |
| | | REM | 68.80 | 35.88 | 29.12 | 32.50 | 49.48 | 60.60 | 40.59 | 50.60 | 84.98 | 16.03 | 13.02 | 14.52 |
| | | AR | 40.53 | 59.47 | 39.82 | 49.64 | 50.27 | 60.14 | 40.13 | 50.13 | 40.75 | 14.12 | 59.72 | 36.92 |
| | 0.8 | EM | 43.20 | 78.72 | 17.55 | 48.14 | 51.24 | 79.75 | 19.77 | 49.76 | 98.13 | 2.08 | 0.88 | 1.48 |
| | | OPS | 70.51 | 35.57 | 25.90 | 30.74 | 50.11 | 80.42 | 20.40 | 50.41 | 95.98 | 0.09 | 16.52 | 8.30 |
| | | LSP | 75.62 | 56.97 | 10.10 | 33.53 | 52.68 | 81.04 | 20.82 | 50.93 | 96.09 | 3.19 | 6.97 | 5.08 |
| | | REM | 58.79 | 72.54 | 13.19 | 42.87 | 56.64 | 79.94 | 19.95 | 49.95 | 90.52 | 8.81 | 13.37 | 11.09 |
| | | AR | 49.66 | 79.83 | 19.93 | 49.88 | 49.37 | 80.69 | 20.68 | 50.69 | 82.35 | 17.20 | 26.16 | 21.68 |
| | Average | | 56.07 | 34.52 | 42.43 | 38.50 | 50.37 | 50.15 | 50.12 | 50.14 | **84.50** | **17.25** | **17.75** | **17.50** |

tacks for comparison. TeCo (Liu et al., 2023a) is an innovative test-time poisoned sample detection method designed for backdoor attacks. It evaluates the consistency of test-time robustness by measuring the extent of severity deviation, which triggers shifts in predictions across different corruptions. Deep k-NN (Peri et al., 2020) is a straightforward, yet highly-effective detection against clean-label backdoor attacks, and exploits the property that adversarial examples have different feature distributions than their clean counterparts in deeper layers of the network. Furthermore, these adversarial features are typically aligned closely to the distribution of the target class.

**Metrics.** To evaluate the performance of availability attacks, we adopt the Half Total Error Rate (HTER) and the F1-score as our evaluation metrics. HTER is formulated as: $\text{HTER} = \frac{\text{FRR+FAR}}{2}$, where $\text{FAR} = \frac{\text{FP}}{\text{FP+TN}}$ and $\text{FRR} = \frac{\text{FN}}{\text{FN+TP}}$. It integrates both the False Rejection Rate (FRR) and the False Acceptance Rate (FAR) to provide a holistic view of the performance. The F1 score is calculated by $\text{F1 score} = \frac{2 \times (\text{precision} \times \text{recall})}{(\text{precision} + \text{recall})}$.

## 4.2 EFFECTIVENESS ON VARIOUS TYPES AND DIFFERENT POISON RATIOS

**Results on CIFAR-10 dataset.** We evaluate the performance of ISR on different unlearnbale examples comprehensively. In terms of the poison ratios, we focus on evaluating at 20%, 40%, 60%, and 80%. The results in Table 1 demonstrate that ISR can successfully identify the trigger samples, particularly when the poison ratio exceeds 20%, as confirmed by the fact that most HTER values below 10%. However, with only a 20% poison ratio, the poisoned samples represent a minority, and additional experiments reveal that a marginal percentage of UEs does not significantly affect testing performance. Therefore, the effectiveness of detection becomes more critical when dealing with larger poison ratios. Notably, on the CIFAR-10 dataset, ISR achieves an average HTER of 0.1750, F1 score of 0.8450, FAR of 0.1725, and FRR of 0.1775. These results showcase ISR's superiority, surpassing the runner-up by roughly 20% in HTER, 30% in F1-score, 17% in FAR, and 25% in FRR. Certainly, there are scenarios in which ISR may not achieve success, particularly when dealing with AutoRegressive (AR) poisons. AR poisons utilize autoregressive perturbations, making them more complex and not linearly separable (Sandoval-Segura et al., 2022). This complexity can make them more challenging to detect, especially when the poison ratios are low. Figure 3 illustrates the performance at each iteration. It is noticeable that once retrieval is integrated and the filtering process continues through several iterations, the method demonstrates improved convergence performance compared to the results preceding retrieval. In summary, our work delivers consistent effectiveness across various types of UEs without using extra knowledge.

There are also some interesting findings about baselines. TeCo, although highly effective in detecting poisoned samples for backdoor attacks, appears to completely fail when dealing with unlearnable

Table 2: Detection performance on CIFAR-100 dataset.

| Dataset | Ratio | Poisons | Deep k-NN (Peri et al., 2020) | | | | TeCo (Liu et al., 2023a) | | | | Our Method | | | |
|---|---|---|---|---|---|---|---|---|---|---|---|---|---|---|
| | | | F1-score↑ | FAR↓ | FRR↓ | HTER↓ | F1-score↑ | FAR↓ | FRR↓ | HTER↓ | F1-score↑ | FAR↓ | FRR↓ | HTER↓ |
| CIFAR-100 | 0.2 | EM | 52.10 | 10.80 | 73.53 | 42.16 | 45.79 | 19.99 | 79.99 | 49.99 | 81.32 | 37.92 | 17.56 | 27.74 |
| | | OPS | 50.83 | 10.16 | 73.56 | 41.86 | 48.05 | 19.89 | 79.91 | 49.90 | 80.88 | 37.79 | 18.41 | 28.10 |
| | | LSP | 45.53 | 17.78 | 78.44 | 48.11 | 47.68 | 19.89 | 79.91 | 49.90 | 79.17 | 79.68 | 19.99 | 49.83 |
| | 0.4 | EM | 73.06 | 1.97 | 39.95 | 20.96 | 48.25 | 40.54 | 60.42 | 50.48 | 93.03 | 4.23 | 8.50 | 6.36 |
| | | OPS | 69.73 | 5.47 | 42.72 | 24.1 | 48.89 | 40.87 | 60.83 | 50.85 | 90.44 | 2.94 | 12.72 | 7.83 |
| | | LSP | 66.88 | 8.33 | 44.72 | 26.52 | 45.91 | 42.64 | 61.78 | 52.21 | 78.58 | 11.66 | 24.53 | 18.09 |
| | 0.6 | EM | 90.27 | 2.94 | 12.93 | 7.93 | 50.78 | 59.90 | 39.91 | 49.90 | 96.86 | 4.03 | 1.67 | 2.85 |
| | | OPS | 68.80 | 3.69 | 43.64 | 23.66 | 49.71 | 60.97 | 40.86 | 50.91 | 92.07 | 1.70 | 15.19 | 8.45 |
| | | LSP | 85.29 | 7.44 | 17.79 | 12.62 | 50.64 | 60.30 | 40.25 | 50.28 | 92.79 | 6.29 | 8.59 | 7.44 |
| | 0.8 | EM | 42.26 | 75.69 | 8.95 | 42.32 | 56.99 | 81.03 | 20.60 | 50.81 | 97.52 | 2.62 | 1.79 | 2.21 |
| | | OPS | 71.18 | 60.31 | 5.25 | 32.78 | 49.53 | 80.96 | 20.93 | 50.94 | 96.05 | 1.97 | 11.44 | 6.71 |
| | | LSP | 71.78 | 59.41 | 4.78 | 32.09 | 56.78 | 77.39 | 18.08 | 47.73 | 97.07 | 1.59 | 8.11 | 4.85 |
| | Average | | 65.64 | 22.00 | 37.19 | 29.59 | 49.92 | 50.36 | 50.29 | 50.33 | **89.65** | **16.04** | **12.38** | **14.21** |

Table 3: Detection performance on 100-class ImageNet subset.

| Dataset | Ratio | Poisons | Deep k-NN (Peri et al., 2020) | | | | TeCo (Liu et al., 2023a) | | | | Our Method | | | |
|---|---|---|---|---|---|---|---|---|---|---|---|---|---|---|
| | | | F1-score↑ | FAR↓ | FRR↓ | HTER↓ | F1-score↑ | FAR↓ | FRR↓ | HTER↓ | F1-score↑ | FAR↓ | FRR↓ | HTER↓ |
| ImageNet-100 | 0.2 | EM | 41.39 | 19.92 | 80.14 | 50.03 | 49.84 | 20.27 | 80.20 | 50.23 | 93.03 | 23.59 | 1.50 | 12.55 |
| | | OPS | 42.31 | 18.24 | 79.05 | 48.64 | 50.32 | 20.82 | 80.08 | 50.85 | 90.08 | 26.62 | 5.31 | 15.97 |
| | | LSP | 49.90 | 6.56 | 72.95 | 39.76 | 50.18 | 17.90 | 78.14 | 48.02 | 88.77 | 22.76 | 9.01 | 15.89 |
| | 0.4 | EM | 47.11 | 40.27 | 59.93 | 50.10 | 48.57 | 40.85 | 60.75 | 50.80 | 94.03 | 11.36 | 1.70 | 6.53 |
| | | OPS | 60.16 | 18.01 | 49.88 | 33.94 | 49.02 | 41.07 | 61.09 | 51.08 | 91.71 | 12.71 | 5.00 | 8.85 |
| | | LSP | 68.56 | 4.15 | 43.62 | 23.88 | 55.13 | 34.48 | 54.75 | 44.61 | 92.03 | 15.35 | 1.60 | 8.47 |
| | 0.6 | EM | 52.92 | 60.34 | 39.84 | 50.09 | 48.99 | 62.05 | 41.72 | 51.89 | 95.76 | 6.33 | 0.05 | 3.43 |
| | | OPS | 77.9 | 23.19 | 21.55 | 23.37 | 48.45 | 61.24 | 41.31 | 51.27 | 94.64 | 4.79 | 6.21 | 5.50 |
| | | LSP | 85.88 | 3.93 | 18.08 | 11.01 | 56.74 | 53.97 | 35.84 | 44.90 | 95.71 | 4.80 | 3.46 | 4.13 |
| | 0.8 | EM | 58.79 | 79.69 | 19.69 | 49.69 | 49.28 | 81.01 | 20.99 | 51.00 | 97.22 | 3.14 | 1.04 | 2.09 |
| | | OPS | 80.36 | 49.21 | 5.60 | 27.4 | 49.47 | 80.70 | 20.69 | 50.70 | 94.72 | 4.61 | 8.33 | 6.47 |
| | | LSP | 44.87 | 74.89 | 7.15 | 41.02 | 57.74 | 76.01 | 17.01 | 46.51 | 96.24 | 3.22 | 6.10 | 4.66 |
| | Average | | 59.18 | 33.20 | 41.46 | 37.41 | 52.23 | 47.50 | 50.23 | 49.32 | **93.66** | **11.53** | **4.11** | **7.88** |

examples. This divergence in performance could be attributed to the differing objectives of backdoor attacks and UEs. Backdoor attacks aim to manipulate the prediction results, shifting them from the source class to the target class after the integration of triggers. On the other hand, unlearnable examples do not necessarily require a prediction shift after the perturbations are introduced. This difference in objectives illuminates the varying performance of TeCo in these diverse situations. Deep k-NN appears to be effective when the number of UEs and clean data is well balanced. However, it seems to fail when dealing with a critical scenario where UEs significantly outnumber the clean data. This indicates that Deep k-NN's efficiency is closely tied to data distribution, particularly when there is a significant imbalance between UEs and clean data.

**Results on CIFAR-100 dataset and 100-class ImageNet subset.** In our experiments on both the 100-class datasets, we select the three most representative unlearnable examples methodologies. As the experimental results are shown in Table 2 and Table 3, our ISR strategy consistently outperforms competing methods in terms of detection and defense against UEs.

## 4.3 ABLATION STUDY

We conduct experiments to show the effectiveness of the proposed detection strategy, comparing it with the method without the retrieval process and without introducing additional classes. The experimental results are shown in Table 4. It can be seen that introducing additional classes and including the filtered clean data in the training process both significantly improve the detection performance. Specifically, the retrieval process has demonstrated its effectiveness, particularly in critical scenarios with a high poison ratio This enhancement is most noticeable in the False Rejection Rate (FRR), a key metric that evaluates the effectiveness of identifying malicious UEs. With the incorporation of the retrieval process, a significant decrease in FRR is observed. Furthermore, through t-SNE, we illustrate the feature distribution of the model, comparing training scenarios without and with the additional clean data (having updated labels $y \in [C, 2C - 1]$), as shown in Figure 4. It is evident that the additional clean data serves to separate the clean data from the UEs in the latent space, thereby improving the detection performance.

Table 4: Ablation study of the proposed method on CIFAR-10 dataset.

| Poisons | Methods / Ratio | 0.4 | | | | 0.6 | | | | 0.8 | | | |
|---|---|---|---|---|---|---|---|---|---|---|---|---|---|
| | | F1-score↑ | FAR↓ | FRR↓ | HTER↓ | F1-score↑ | FAR↓ | FRR↓ | HTER↓ | F1-score↑ | FAR↓ | FRR↓ | HTER↓ |
| EM | w/o retrive | **94.52** | 4.17 | **6.25** | **5.21** | 91.44 | **3.01** | 17.41 | 8.85 | 97.05 | **1.40** | 8.82 | 5.11 |
| | w/o additional classes | 82.92 | 29.20 | 2.20 | 15.70 | 78.53 | 26.27 | 0.69 | 13.48 | 94.56 | 6.05 | 2.02 | 4.03 |
| | Our Method | 88.88 | **0.10** | 15.60 | 7.85 | **95.98** | 4.06 | **3.93** | **4.00** | **98.13** | 2.08 | **0.88** | **1.48** |
| OPS | w/o retrive | **94.33** | 16.72 | **7.86** | **4.77** | 94.48 | 3.85 | **7.90** | 5.88 | 94.84 | 0.60 | 19.37 | 9.99 |
| | w/o additional classes | 83.90 | 26.59 | 5.17 | 15.88 | 87.75 | 14.96 | 6.45 | 10.70 | 92.33 | 6.70 | 12.53 | 9.62 |
| | Our Method | 92.96 | **0.55** | 10.02 | 5.41 | **95.25** | **2.37** | 8.03 | **5.20** | **95.98** | **0.09** | 16.52 | 8.30 |

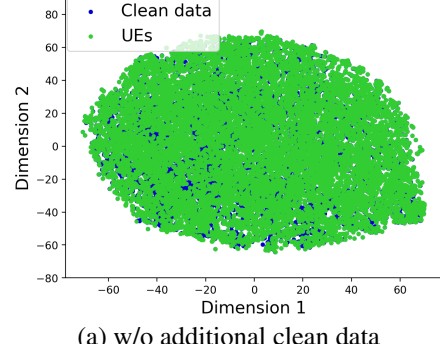

(a) w/o additional clean data

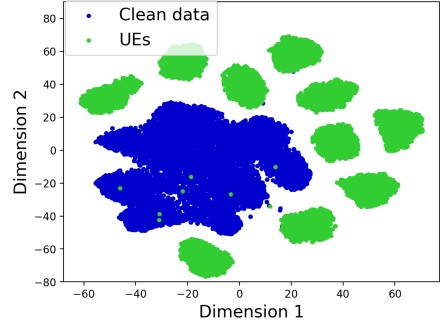

(b) w/ additional clean data

Figure 4: t-SNE visualizations on CIFAR-10, comparing models trained without and with additional clean data, where the labels for the additional clean data are updated to $y \in [C, 2C - 1]$. Note that UEs are generated by EM, and the shared training data consists of 50% UEs and 50% clean data.

## 4.4 DETECTION FOR PURIFICATION

In this section, we demonstrate the effectiveness of our approach by conducting experiments dedicated to defending against UEs while adhering to detection-defense principles. It is worth noting that we choose Diffpure (Dolatabadi et al., 2023), which utilizes a diffusion model for purification, as the defense method in these experiments. As evident in Table 5, our proposed detection methods prove highly effective

Table 5: Test Accuracy (%) of models trained on CIFAR-10 with various defensive methods. Dp denote Diffpure.

| Ratio | Defensive method | EM | OPS | LSP | REM | AR | Mean |
|---|---|---|---|---|---|---|---|
| 0.2 | w/o | 94.27 | 93.91 | 94.61 | 94.06 | 94.24 | **94.21** |
| | Dp on all samples | 89.61 | 72.64 | 89.67 | 89.92 | 89.93 | 86.35 |
| | Dp on detected samples | 94.20 | 94.02 | 94.21 | 93.99 | 94.09 | 94.10 |
| 0.4 | w/o | 93.42 | 92.63 | 93.28 | 93.18 | 93.23 | 93.14 |
| | Dp on all samples | 89.44 | 71.53 | 89.60 | 89.67 | 89.81 | 86.01 |
| | Dp on detected samples | 93.63 | 93.21 | 93.10 | 93.28 | 93.30 | **93.30** |
| 0.6 | w/o | 91.86 | 91.76 | 91.89 | 91.71 | 91.68 | 91.78 |
| | Dp on all samples | 89.45 | 69.55 | 89.80 | 90.02 | 89.39 | 85.64 |
| | Dp on detected samples | 92.90 | 91.83 | 92.83 | 91.51 | 91.79 | **92.17** |
| 0.8 | w/o | 87.94 | 87.17 | 88.19 | 86.58 | 87.51 | 87.47 |
| | Dp on all samples | 89.86 | 70.39 | 89.73 | 89.87 | 89.70 | 85.91 |
| | Dp on detected samples | 91.48 | 89.87 | 90.22 | 91.52 | 89.97 | **90.61** |

in improving defense performance, particularly in scenarios with significant poison ratios. Note that all experiments were conducted on CIFAR-10 dataset.

## 5 CONCLUSION

In this paper, we present Iterative Self Regression (ISR), a robust and efficient detection technique designed to identify a wide range of visually imperceptible Unlearnable Examples (UEs). Our approach leverages a critical property: Unlearnable Examples (UEs) tend to be acquired by the classifier more rapidly than clean data when training on a partially poisoned dataset. This difference inspires us to introduce an iterative algorithm for data separation. Moreover, we highlight that distinguishing between clean and poisoned samples is more effective. Through the introduction of the additional classes and the adoption of iterative refinement, our proposed approach achieves improved effectiveness in classifying these two sample types. Extensive experiments conclusively demonstrate that our method outperforms state-of-the-art detection approaches when challenged with various types of attacks, datasets, and poisoning ratios.

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
