# OpenReview forum: "Screening Unlearnable Examples via Iterative Self Regression"
_ICLR.cc/2024/Conference — ICLR 2024 Conference Withdrawn Submission_

### Official Review · Reviewer_6SZH · 2023-10-29

**Soundness:** 2 fair
**Presentation:** 1 poor
**Contribution:** 1 poor
**Rating:** 1
**Confidence:** 5

**Summary:**

This paper considers the problem of detecting unlearnable examples on a mixed dataset (i.e., consists of clean data and UEs). The authors propose an iterative self regression algorithm and design relevant experiments to show its effectiveness.

**Strengths:**

The iterative self regression algorithm seems interesting.

**Weaknesses:**

**[Availability attacks]**: the claims regarding availability attacks are not accurate and correct throughout the paper. An example is:
> While most of the literature focuses on the malicious use of poisoning attacks, availability attacks
are employed as a protective measure against unauthorized model training.

The authors should be aware of the definition of availability attacks, namely a type of data poisoning attack that aims to decrease the overall test accuracy. There are generally two types of attacks in this category: indiscriminate attacks that inject a small portion of poisoned data (for attack purposes, e.g., Biggio et al., 2012, Koh et al., 2022, etc) and unlearnable examples that perturb the entire training set (for protection purposes, e.g., Huang et al., 2020, etc). Thus availability attack $\neq$ unlearnable examples. I suggest the authors study the literature well, cite necessary papers in this field, and then present proper claims.

**[Questionable Problem of Interest]**: the authors consider a scenario where there exists a dataset that consists of both unlearnable examples ($\mathcal{D}_u$) and clean data ($\mathcal{D}_c$) (with a ratio of 50%/50%). This setting is itself problematic to me due to:
- Unlearnable examples are hardly a threat in this case: in Table 2 of Huang et al., 2020, the authors show that even when the percentage of unlearnable examples is 60% in a mixed dataset, the incurred test accuracy drop is only 2.13%. In other words, the success of unlearnable examples depends on the assumption that the entire training set is to be perturbed.
-  Thus I would say detecting unlearnable examples is pointless in a mixed dataset. In fact, the authors even claim that:
> As a result, it is difficult for data users to determine whether the dataset is normal because the trained model appears to have reasonable performance. Training on these unlearnable examples may not fully leverage the model’s potency, compromising testing performance and wasting computational resources and time.

   If the performance is reasonable, why do we care if the dataset is normal? In other words, do we care to defend against a data poisoning attack that is poor? Additionally, these two sentences are contradicting each other and are not reasonable to me.

- In my opinion, the proposed work seems to be more suitable for the problem of indiscriminate attacks (mentioned above), where adding a small portion of poisoned data into a clean training set could be an actual threat and a detection mechanism is needed.

**[Key intuition of the paper]**: to motivate the proposed method, the authors claim that:
> This suggests that when the model is evaluated on previously unseen UEs, it often demonstrates superior accuracy compared to its performance on clean samples.

This critical insight (further presented in Sections 3.1, 3.2, and Figure 1) is based on the assumption that UEs appear in the test set, which requires the attacker to also have access to the test set.  I believe this attack capability is not considered in the threat model of UEs and I don't understand how this key intuition would provide any information regarding detection.

**[Table 5]**: one purpose of detection is to leverage the performance drop by removing the unlearnable examples. However, I don't see the relevant results in Table 5. Additionally, after the purification of detected samples, the accuracy increase is tiny (and even an accuracy drop of 0.11% for ratio=0.2). To me, the results suggest that the detection and purification process is an expensive trade-off and in practice utilizing this might be pointless, especially since the accuracy without any defense is relatively reasonable.

**[Presentation]**: reading this paper is a difficult process for me as (1) many of the sentences are ambiguous; (2) lack of proper captions on figures and tables; (3) some unnecessary repetitions.

**In summary**,  I think this paper could be largely improved in terms of literature review and presentation. More importantly, the authors should revisit the problem setting and evaluate if it is valuable to study.

**Questions:**

I don't have any other questions. The authors may address my comments in the weakness section.

---

### Official Review · Reviewer_Ymu2 · 2023-10-30

**Soundness:** 2 fair
**Presentation:** 2 fair
**Contribution:** 2 fair
**Rating:** 3
**Confidence:** 4

**Summary:**

This paper proposes to detect Unlearnable Examples (UEs), one type of availability attacks, using the accuracy gap between the training and validation subsets. It then identifies those incorrectly predicted samples into an extended subset with class labels reassigned to $y+C$. Training on the mixed dataset further enlarges the accuracy gap and makes the UEs trapped into the first $C$ classes (whilst the clean samples were gradually moved into the extended $C$ classes). The idea was tested on 5 UE generation methods under large poisoning rates ranging from 20% to 80%.

**Strengths:**

1. The use of the accuracy gap to identify and remove poisoned samples is quite interesting.

2. The proposed method seems to be quite effective against the rest attacks.

3. The reason why adding new classes can help detect UEs is visually explained.

**Weaknesses:**

1. The main detection algorithm is hard to understand, there are many places unclear. For instance, what is iterative self-regression? How many epochs/iterations are needed to detect all poisoned samples?

2. UEs usually assume a 100% poisoning rate, from the perspective of data protection. It is not clear how to address this assumption using the proposed method.

3. There are claims used interchangeably for backdoor samples and UEs, yet only UEs were tested in the experiments. Should the authors take backdoor samples as one type of poisoned data and test the proposed method on them as well?

4. For the competing methods, the authors argued that they are the first UE detection method. However, the reviewer believes that all backdoor sample detection methods can directly be applied here to detect UEs.

5. The authors explained that their method "incorporated additional classes and iterative refinement,", yet it is not clear where is the refinement. In other words, what has been refined and how?

6. The observation that poisoned samples learn faster was systematically studied in [1] for backdoored samples. What is the difference between the observation made in this paper and that in [1]?


[1] Li, Yige, et al. "Reconstructive Neuron Pruning for Backdoor Defense." ICML, 2023.

**Questions:**

1. How does the proposed method deal with a 100% poison rate?

2. In the 2nd last para. of Section 3.2, Eq. (4), it is not clear which samples have their class labeled extended? And missing details of the adjustment of the network architecture as now has $2C$ classes. Is it practical for large-scale datasets with more than 1k classes like ImageNet?

3. It is hard to understand "This result is intuitive, as introducing clean data with additional class labels can lead to these clean data points being misclassified as y + C, thereby amplifying the difference between UEs and clean data."

4. Does the proposed method need a warmup process as otherwise poisoned samples could be misidentified as clean ones at an early training stage?

5. Is the proposed method robust to adaptive attacks like UEs that demonstrate a long time misclassification before being correctly classified?

6. In the algorithm, what does the "retrieved" mean? it is not easy to understand the role of "N_{thre}".

**Details Of Ethics Concerns:**

No ethics concerns.

---

### Official Review · Reviewer_cckS · 2023-11-03

**Soundness:** 2 fair
**Presentation:** 3 good
**Contribution:** 3 good
**Rating:** 5
**Confidence:** 4

**Summary:**

This work introduces a method to screen datasets that include unlearnable examples (UEs). Given the phenomenon that learning models adapt more rapidly to UEs than to clean samples, the authors evaluate the model on previously unseen UEs to detect poisoning samples in the dataset. It is observed that the trained model has a much higher accuracy on UEs than on clean samples. The authors compare the proposed approach with baselines adopted from backdoor detection. In addition, ablation results provide evidence on the necessity of the retrieval module and the additional class module.

**Strengths:**

- This paper is well-written and easy to follow. The idea of detecting and filtering unlearnable examples in a mixed dataset is interesting.

- Especially, the proposed approach relies on neither an external dataset nor an external model, which makes the detection approach more practical.

**Weaknesses:**

- The influence of **early stopping** can be further clarified. On a relatively simple dataset, like CIFAR-10, the training loss of DNNs converged rapidly, which may strongly influence the detection performance. Because the difference between the clean samples and UEs will be less substantial. As a key component, it would be great if the authors could provide more details on early stopping, including answering the following questions: is it a necessary step for the detection? How to set the early stopping criteria? What is the influence of the early stopping criteria on the detection performace?
- Missing comparison to the **detection on clean training data**. Practically, for the dataset with a low poisoning rate, given the nature of the proposed approach, it might be used blindly without suspecting the existence of the UEs. Experimental results on a clean dataset or dataset poisoned with less than 20% of UEs would be interesting. It can show the influence of the screening method on the utility of the model. I would anticipate that there might be a slight accuracy drop in these cases.
- **Edge cases of UEs** are also interesting to discuss further. Why can some UEs not be detected by the proposed approach? Is it because the model cannot converge rapidly on them?
- Another question related to the corner cases is the **adaptive UEs**. For the UEs that cannot be detected, will it be possible to craft such UEs directly? Especially when the adversary is aware that the detection is based on the convergence difference between clean samples and UEs, is it possible to develop a poisoning method that bypasses the proposed defense? It would be great if the authors could discuss it to provide further evidence on the robustness of the detection.

minors:
- In algorithm 1, there is a typo retrived -> retrieved. Should the indicator before the if be y^{hat}_{i} != y_i? Will the retrieval step only be done once in total?

**Questions:**

See weakness part.

---

### Official Review · Reviewer_cGpA · 2023-11-05

**Soundness:** 3 good
**Presentation:** 2 fair
**Contribution:** 3 good
**Rating:** 5
**Confidence:** 4

**Summary:**

This paper studies an interesting problem of filtering the poisoned samples from the training dataset, by using an iterative self regression scheme. The proposed framework is based on a finding that the model often learns from the poisoned samples quicker than the clean data samples. Based on the experimental results, the proposed method is effective on selecting clean samples from a training dataset.

**Strengths:**

1. The studied problem is interesting, since the training data in real-world is often mixed with unclean data samples;
2. The logic and intuition of the proposed method are simple yet effective, and the tease experiment also makes it convincing;
3. Based on the results, the proposed method boosts the performance of detection.

**Weaknesses:**

1. The figures might need more illustrations, for example, there could be more clarifications on the figures in the captions;
2. For the experiment in Figure 1 and Figure 2, while I understand the purpose is to show the model learns the features of UEs quicker than clean data samples, it is a red-flag to use test dataset during the training and multiple times;
3. For the experiments, ResNet-18 is not a popular choice for those tasks;
4. For the experiments, the current datasets are somewhat limited, maybe more tasks can be considered;
5. For the methods for generating UEs, it is more comprehensive to use different perturbation scales rather than pick one for each.

**Questions:**

Please refer to the previous sections. Generally, I think the paper will benefit from conducting more experiments.